

# Sperm removal during copulation confirmed in the oldest extant damselfly, *Hemiphlebia mirabilis*

Adolfo Cordero-Rivera

ECOEVO Lab, Departamento de Ecoloxía e Bioloxía Animal, Universidade de Vigo, Pontevedra, Galiza, Spain

## ABSTRACT

Postcopulatory sexual selection may favour mechanisms to reduce sperm competition, like physical sperm removal by males. To investigate the origin of sperm removal, I studied the reproductive behaviour and mechanisms of sperm competition in the only living member of the oldest damselfly family, *Hemiphlebia mirabilis*, one species that was considered extinct in the 1980s. This species displays scramble competition behaviour. Males search for females with short flights and both sexes exhibit a conspicuous "abdominal flicking". This behaviour is used by males during an elaborate precopulatory courtship, unique among Odonata. Females use a similar display to reject male attempts to form tandem, but eventually signal receptivity by a particular body position. Males immobilise females during courtship using their legs, which, contrarily to other damselflies, never autotomise. Copulation is short (range 4.1–18.7 min), and occurs in two sequential stages. In the first stage, males remove part of the stored sperm, and inseminate during the second stage, at the end of mating. The male genital ligula matches the size and form of female genitalia, and ends by two horns covered by back-oriented spines. The volume of sperm in females before copulation was 2.7 times larger than the volume stored in females whose copulation was interrupted at the end of stage I, indicative of a significant sperm removal. These results point out that sperm removal is an old character in the evolution of odonates, possibly dating back to the Permian.

## INTRODUCTION

The relevance of Sexual Selection in shaping reproductive behaviour is well established (*Andersson, 1994*), and in particular its effects on the evolution of primary (*Leonard & Córdoba-Aguilar, 2010*) and secondary sexual characters (*Clutton-Brock, 2007*). Our understanding of sexual selection processes is based on the conceptual framework that differentiates between pre-copulatory and post-copulatory forces and mechanisms (*Eberhard, 1996*), and the traditional differentiation of intra- and inter-sexual selection, which can be dated back to the original texts of *Darwin (1871)*. The origin and maintenance of many bizarre behaviours is explained by sexual selection theory, even if such behaviours may not be favoured by natural selection. Males and females do not necessarily have the same interests, and conflicts are expected (*Parker, 1979*; *Arnqvist, 2004*), which are

Corresponding author
Adolfo Cordero-Rivera,
adolfo.cordero@uvigo.es

particularly evident in species whose females store sperm for long periods. Animal genitalia are therefore under intense sexual selection (*Eberhard, 1985*), and this may drive coevolutionary arm races between sexes (e.g., *Sánchez & Cordero, 2014*).

Several mechanisms explain the evolution of male reproductive strategies, both to increase paternity when competing with other males, and to reduce the success of other males in the context of sperm competition (for a review, see *Cordero Rivera & Córdoba-Aguilar, 2016*). Perhaps the most surprising adaptation is the ability to physically remove the sperm from rivals, stored inside the female, using the intromittent organ. In a seminal paper, *Waage (1979)* demonstrated, for the first time, that male damselflies are able to use their genital ligula to trap sperm from previous mates, and remove them during copulation, so that paternity success of the last male is usually near 100% in the first clutch laid after copulation (e.g., *Cordero & Miller, 1992*). Since Waage's discovery, the mechanism of sperm removal by males has been demonstrated in several animal groups, including Dermaptera (*Kamimura, 2005*), Orthoptera (*Ono, Siva-Jothy & Kato, 1989*; *Von Helversen & Von Helversen, 1991*) and Coleoptera (*Yokoi, 1990*) among the insects, but also in Crustaceans (*Galeotti et al., 2008*) and Cephalopods (*Wada et al., 2010*). This is therefore a widespread phenomenon in animals, which has apparently evolved several times, but we have little information about its origin in any animal group.

Although Odonates are well known model species for studies of sexual selection and evolutionary biology in general (*Córdoba-Aguilar, 2008*), most research has been confined to a few families (*Cordero-Rivera & Córdoba-Aguilar, 2010*). This taxonomic bias might induce wrong interpretations about the generality of some behaviours. We do not have fossil information to track the evolution of significant behaviours, like the ability to remove sperm from previous mates when a male copulates with a female. An alternative is to study behaviour on a phylogenetic perspective. Given its basal position in the Zygoptera (*Dumont, Vierstraete & Vanfleteren, 2010*), *Hemiphlebia mirabilis*, the only living member of the Hemiphlebiidae, is a priority taxon in this context. This family dates back to the late Jurassic at least (*Lak et al., 2009*), and *H. mirabilis* has been considered a "living-fossil", a survival from the Permian (*Fraser, 1955*).

The goals of this study were therefore two fold. First, I wanted to observe and describe the reproductive behaviour of this species, that has been up to now unknown. *Fraser (1955)* published a description of the male intromittent organ (genital ligula), noting that it had two flagella or horns similar to those of other species known to use these structures to remove sperm (*Córdoba-Aguilar & Cordero-Rivera, 2008*). However, female internal organs remain unknown. Given that female structures are the "arena" where postcopulatory sexual selection takes place, their study is crucial. Therefore, my second goal was to study genital morphology of both sexes and sperm competition mechanisms. If *H. mirabilis* males displace sperm, this would suggest that sperm displacement and the dual function of male genitalia (*Waage, 1979*) is an old character in Odonates.

## METHODS

Most observations and experiments were done at Long Swamp, a large freshwater system in the Discovery Bay Coastal Park, near to Nelson (Victoria, Australia), between 17 November

and 7 December 2013. Population density of *H. mirabilis* was very high during the field work (*Cordero-Rivera, 2016*). A second population, inhabiting Ming Ming swamp in Grampians National Park, was visited for further observations.

Animals included in this study were individually marked. Having a marked population is a convenient fact for demographical and behavioural studies (*Cordero-Rivera & Stoks, 2008*), because individually marked specimens are best for focal observations. Marking was done with a permanent black ink pen (Faber-Castell Multimark 1525 S) on the external side of the right hind wing (see Figs. 2 and 3).

Copulatory behaviour was rarely observed. For instance, I observed one mating on 19 and 25 November and two on 26 November. Therefore, I decided to try to increase inter-individual encounters by using a mosquito net as an outdoor insectary, where a set of marked specimens were introduced. The insectary was used to elicit mating behaviour on days 29, 30 November and 1 and 3 December, between 11 and 16–17 h. At the end of the observations, the insectary was removed and remaining individuals released. I observed a total of 28 copulations, of which 11 were forcibly interrupted at different times to study sperm competition. Three pairs were preserved in copula (at the end of stage I; see 'Results') and dissected to test whether the genital ligula can physically remove sperm. Furthermore, seven apparently mature females were collected to estimate the sperm volume of pre-copula females. The pair, or sometimes only the female, was immediately preserved in 70% ethanol for further analyses. One thermometer was placed in the shade of a shrub at about 1.5 m over water. Temperature was recorded about once every hour.

In the laboratory, females were dissected and the sperm storage organs extracted. Two postcopula females had no sperm in their *bursa copulatrix*. This fact might be due to failure in insemination (*Sánchez-Guillén, Wellenreuther & Cordero-Rivera, 2011*), and these females were excluded from sperm volume analyses. Furthermore, two females collected alone (pre-copula) and also two of the group interrupted during the stage I, had no sperm. These were likely unmated or mating for the first time (and interrupted before insemination), and were also excluded from sperm volume estimations. The sperm storage organs of a further female were accidentally damaged during dissection and could not be analysed. Final sample sizes were therefore five pre-copula females, seven after copulation and six interrupted at the end of stage I. The volume of sperm was estimated following standard protocols (*Cordero & Miller, 1992*), using the software ImageJ (http://imagej.nih.gov/ij/) to measure the area of the sperm mass from microscope pictures. Male secondary genitalia were dissected and observed under Scanning Electron Microscope (SEM). Permits to collect odonates were issued by the Victorian Department of Environment and Primary Industries (permit number 10006907).

Mean values are presented with their standard error (SE) and sample size: mean $\pm$ SE (N). Statistical analyses were performed with xlStat 2016 (www.xlstat.com) and Genstat 17 (http://www.vsni.co.uk). Sperm volumes were analysed by ANOVA, and groups were compared using pre-copula females as the control, by means of a Dunnett test.

## RESULTS

### Reproductive behaviour

*Hemiphlebia mirabilis* showed limited flying activity, and remained perched most of the time. Males were not territorial, and the mating system was found to be scramble competition. The most conspicuous behaviour was abdominal flicking, which occurred continuously over the day, was performed by both sexes, and has been studied elsewhere (*Cordero-Rivera, 2016*).

Mating behaviour was very infrequent. Over the study period, I observed 28 copulations, but only eight of them outside the insectary. Mating behaviour was found to be similar to that of Coenagrionidae, but with several peculiarities. When a male detected a potential partner, he made a fast flight, similar to a feeding flight, and grasped the female by her wings with his legs (Fig. 1A). Occasionally, males tried to achieve tandem with other males (even of other species), but this behaviour lasted only a few seconds. Some females curled their abdomen upwards when immobilised by males, and tried to dislodge the male, in a clear refusal behaviour (Fig. 2), which was sometimes successful. In the insectary, these tandems that did not end in copula lasted $8.4 \pm 2.4$ (4) min.

Receptive females remained motionless, and adopted a characteristic position with their abdomen curved downwards in the junction between the first and second abdominal segments, and upwards between third and fourth segments. The position recalls a capital Z (Figs. 1B–1F and 2C). After a variable time motionless, males started to make their characteristic abdominal flicking, in a clear courtship behaviour (see the sequence in Figs. 1B–1F, and also Video 1). Then, males curved their abdomen upwards and grasped the females's prothorax with their abdominal appendages (Fig. 2C). The average time between female capture and tandem formation was $6.20 \pm 1.17$ (19) min, with a range from 1.67 to 23.0 min. During this time, males used their abdominal flicking display to court the females, with motionless periods intercalated. After grasping the female with his anal appendages, the male opened its legs widely and released the female's wings (Video 1).

After tandem formation, males performed sperm translocation, from the 9th to the 2nd abdominal segment (Fig. 3), which lasted on average $3.5 \pm 0.24$ (26) s (range 1–6 s; Video 1). This behaviour always preceded copulation, contrarily to what was previously reported (*Sant & New, 1988*). Copulation started immediately after sperm translocation. Copulatory movements (see Fig. 4) were similar to those in Coenagrionidae, and had two clear phases, which match the description of stage I and II of *Enallagma cyathigerum* (*Miller & Miller, 1981*).

Copulation lasted $10.53 \pm 1.17$ (14) minutes (range 4.1–18.7 min), and its duration was not significantly related to time of day (Fig. 5) or air temperature (Fig. 6) (multiple regression analysis with copula duration as the response variable and time of day and temperature as explanatory variables; time effect: $-0.0053 \pm 0.0105$, $t_8 = -0.51$, $p = 0.624$; temperature effect: $0.0008 \pm 0.0006$, $t_8 = 1.32$, $p = 0.224$). The analysis of the duration of courtship and precopulatory tandem also suggested that time of day and temperature have no effect ($p$-values $> 0.10$). Most of the variation in copulation duration was due to stage I, which lasted $9.96 \pm 1.30$ (12) min. Stage II was of short duration and showed little

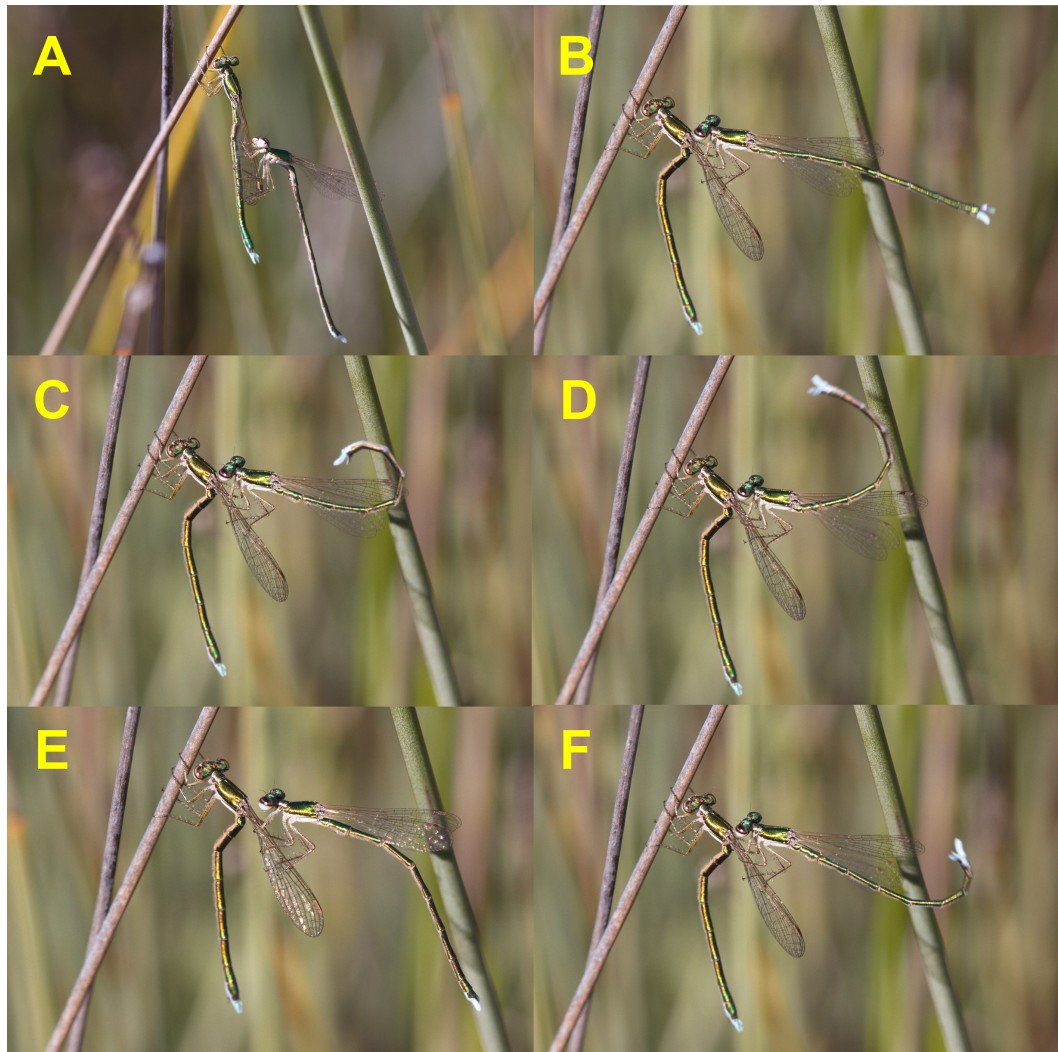

**Figure 1 Courtship behaviour of male *H. mirabilis*.** Males use the "flicking" display characteristic of this species. Note the "Z" position of female abdomen, which signals receptivity. Males repeatedly curl their abdomen up and downwards during this display. See also Video 1.

variability in duration (1.08 ± 0.11 (13) min). At the end of copulation, all males flew off immediately leaving the female alone, but usually perched nearby. Females either flew ($N = 5$) or remained perched for a short time ($N = 4$). One female could be closely observed for about 2 min after copula. She apparently did not expel sperm (*Córdoba-Aguilar, 2006*), but made conspicuous movements of her external genitalia.

I did not observe a single female laying eggs. Oviposition does not take place in tandem after mating, so it should be performed by females alone. On 25 November I collected 10 females apparently mature and put them in plastic containers with humid filter paper, a method that elicits oviposition behaviour in many zygopterans (*Van Gossum, Sánchez-Guillén & Cordero-Rivera, 2003*). They were retained for two hours on a shaded spot, and afterwards were released, but none laid eggs. A second attempt to obtain eggs was done with

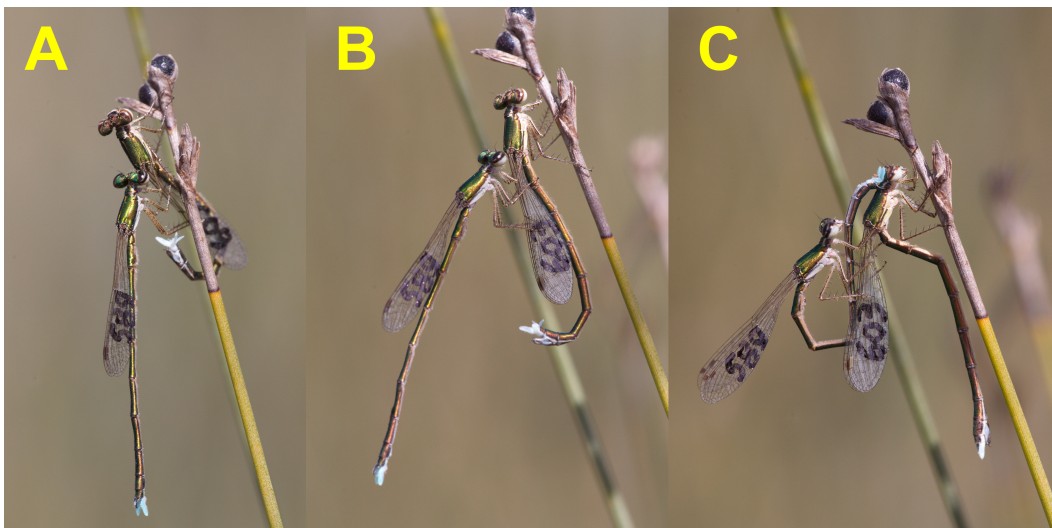

**Figure 2** **Female refusal behaviour, using the abdomen to try to dislodge the male (A and B), and the start of precopulatory tandem (C), once female shows signs of receptivity.** Female acceptance of copulation is indicated by the "Z" position of her abdomen.

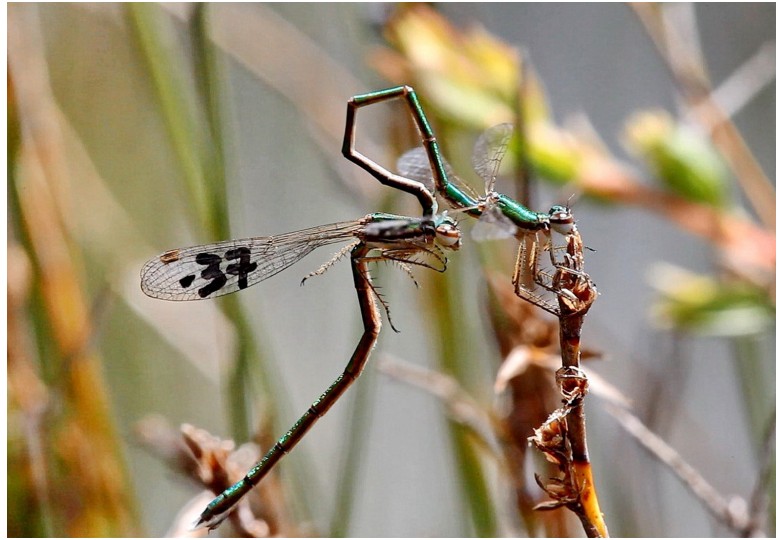

**Figure 3** **Intra-male sperm translocation behaviour.** All matings observed ($N = 28$) were preceded by this behaviour.

22 females from the population at Ming Ming Swamp, which were fully mature judging by their enlarged abdomen. Females were maintained with humid filter paper from 12:15 to 15:30 h, but once more, no oviposition was observed. Five females were retained and maintained overnight with humid filter paper and a piece of vegetation, but yet again, no eggs were laid.

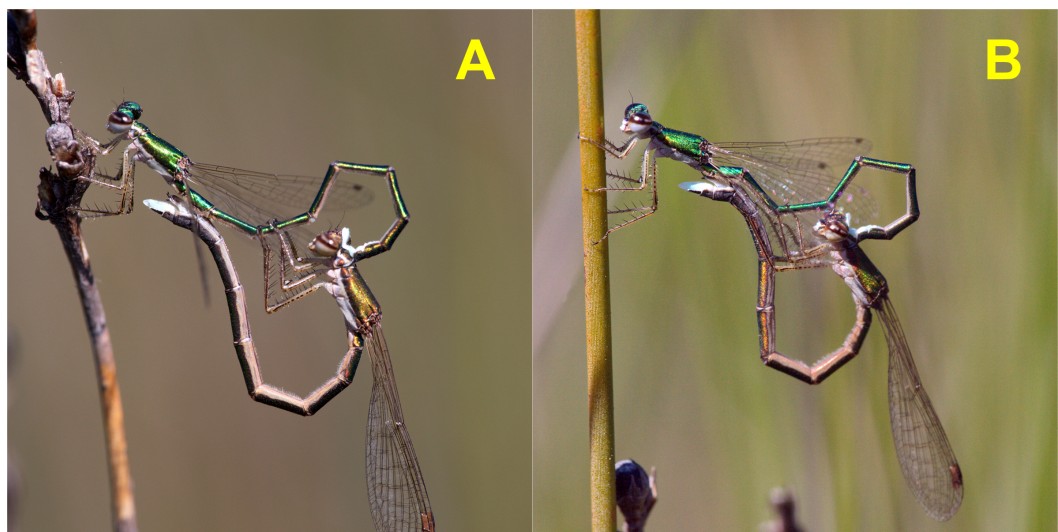

**Figure 4** **Copulatory phases in *H. mirabilis* match the description of stage I (A) and stage II (B) of *Miller & Miller (1981)*.** Stage I is involved in rivals' sperm removal and insemination takes place during stage II.

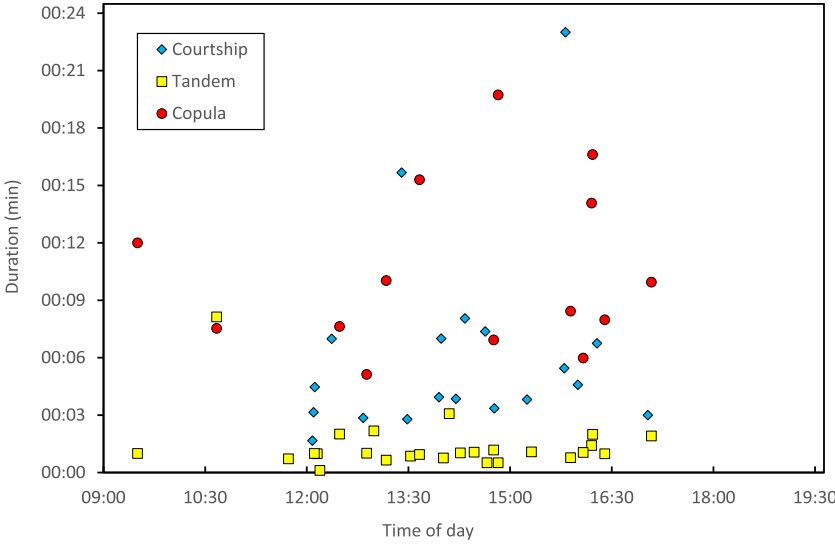

**Figure 5** **Relationship between time of day and duration of reproductive behaviours of *H. mirabilis*.** Courtship refers to the time between female capture and tandem formation. Tandem indicates the time the pair remained in tandem before copulation. Finally, copulation refers to the time between the start and the end of the copulatory wheel.

## Anatomy of genitalia and sperm competition

Figure 7 shows the anatomy of male genitalia under SEM, and female genitalia under an optical microscope. The genital ligula measures about 2 mm in length, and ends in a flexible tip, culminated distally by two dorsal horns or cornua (Figs. 7A and 7D). These are covered by backwards oriented spinules (Fig. 7F). The genital ligula is also covered by small spinules on both sides (Fig. 7E), and by a group of larger spinules on the ventral

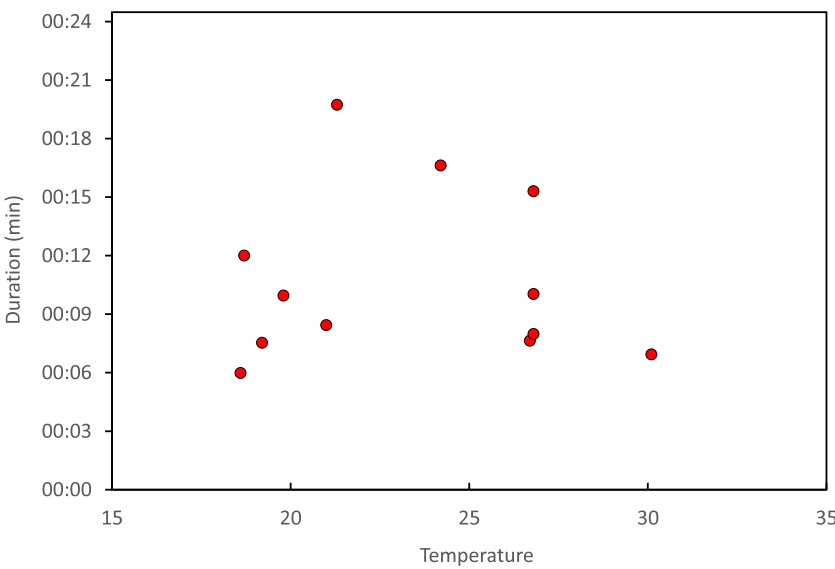

**Figure 6  Relationship between air temperature and copulation duration in *H. mirabilis*.**

median part (Fig. 7C). Female genitalia show two chitinised plates with embedded sensillae (Fig. 7B), and a large *bursa copulatrix*, full of sperm in postcopula females. There is no spermatheca, although in some specimens the *bursa* seems to be divided into two parts as in Fig. 7B. The size and structure of male genitalia compared to female *bursa copulatrix*, as well as the presence of spinules, both suggest that males can remove sperm during the stage I of copulation. Three females, preserved in copula, were dissected. The genital ligula was situated between the chitinised plates in all cases, and in one case the distal part was found inside the bursa, showing that physical removal of sperm is possible (the bursa was empty, likely because the female was unmated).

Pre- and postcopula females had their bursa full of sperm, whereas it was almost empty in females interrupted at the end of stage I (Fig. 8; ANOVA, $F_{2,15} = 11.12$, $p < 0.001$). The volume of sperm in pre-copula females was 2.7 times greater than the volume stored in females whose copulation was interrupted at the end of stage I (difference = 0.007, Dunnett test (two-sided) $p = 0.020$), indicating a significant sperm removal. The volumes of sperm stored by pre- and postcopula females were not significantly different (difference $= -0.003$; $p = 0.293$; Fig. 8).

## DISCUSSION

The reproductive behaviour of *H. mirabilis* is unique from several points of view. Both sexes frequently perform abdominal flicking, particularly after flights, even in the absence of conspecifics (*Cordero-Rivera, 2016*). The results of this study indicate that this abdominal display, which is the most conspicuous behaviour of *H. mirabilis* (*Sant & New, 1988*), is also part of the courtship (Fig. 1). There have been suggestions in the literature indicating that males use the curling of the abdomen to display to and attract females (*Tillyard, 1913*), and that females respond to male abdominal flicking by performing the same display

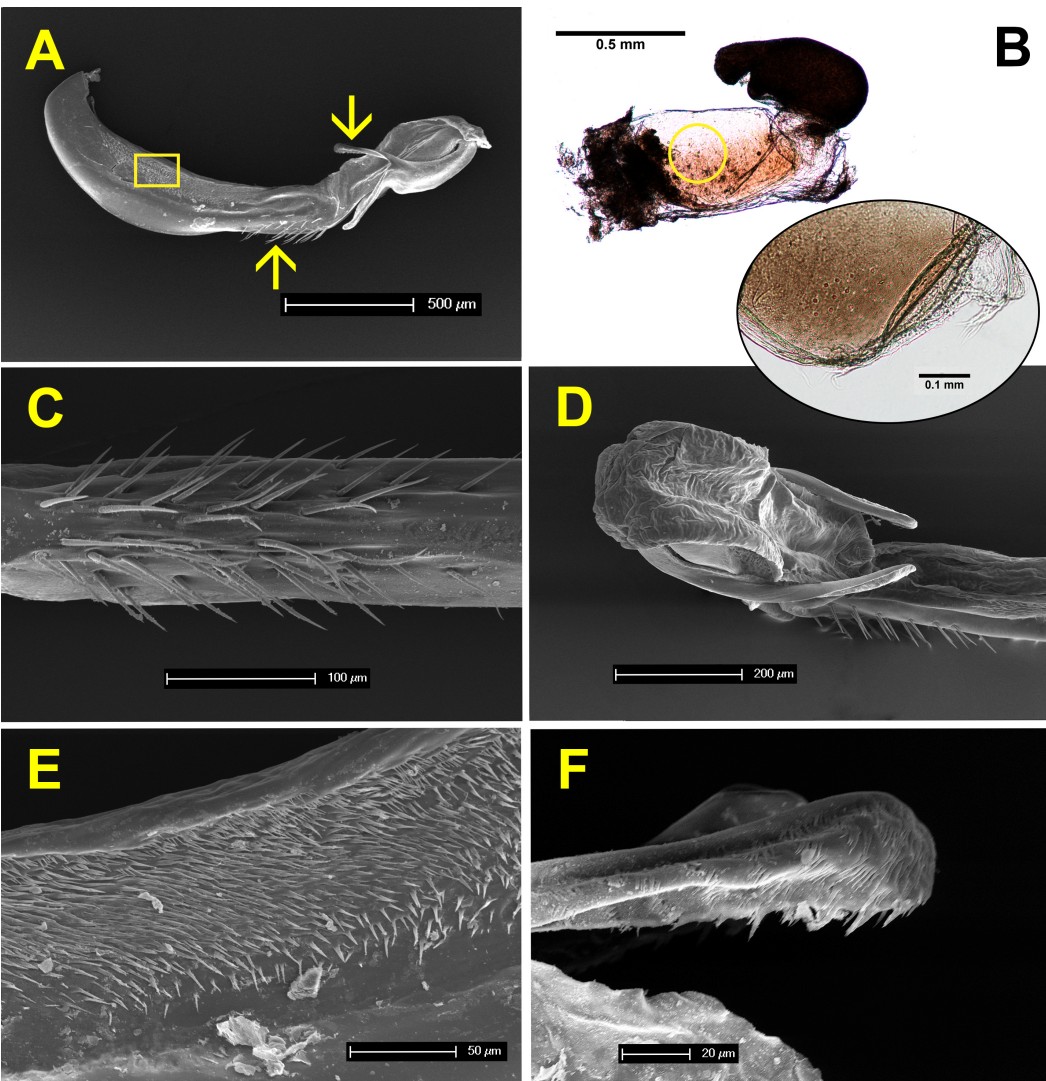

**Figure 7** **Male and female genitalia of *H. mirabilis*.** (A) Lateral view of genital ligula and (B) female vagina and *bursa copulatrix*, both at the same scale, and oriented with the dorsal part above. The insert in (B) shows a detail of the genital valves of the female, which are provided with mechanical sensilla, situated in the area indicated by the circle. Further details of male genital ligula are shown. (C) Ventral view of the spines, in the region indicated by the upwards arrow in (A). (D) Dorso-lateral view of the tip of the genital ligula. (E) Detail of the spinules of the genital ligula in the region indicated by the rectangle in (A). (F) Lateral view of the genital ligula distal horns, with backward-directed spinules, whose position is indicated by the downward arrow in (A). The image in (B) has been edited to remove dust.

(*Davies, 1985*). My observations nevertheless do not support this. Although to some extent both males and females show this behaviour to conspecifics which are also displaying, females can perform abdominal flicking even more actively than males when they are alone and undisturbed (up to 172 times in 10 min compared to 119 times in males; *Cordero-Rivera, 2016*). Abdominal displays may also help in intraspecific recognition, or be a "receptivity" signal, but when males grasp females, they serve as courtship displays. Courtship lasted a maximum of 23 min, which is a substantial amount of time, and is likely to be energetically
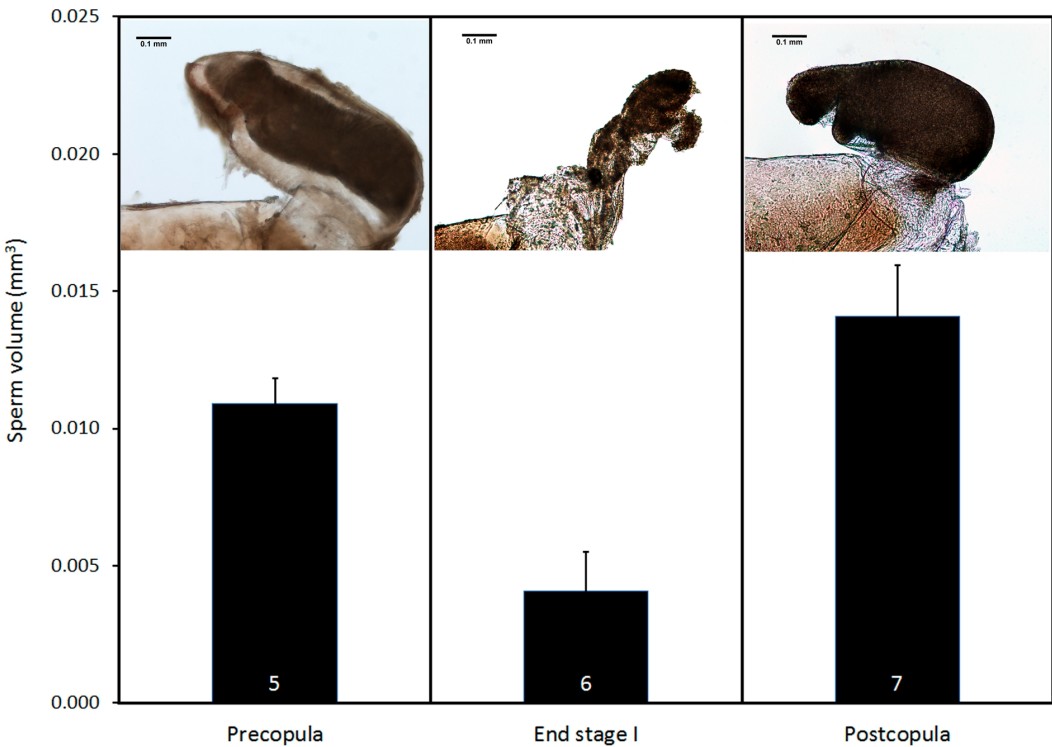

**Figure 8** **Volume of sperm (mean + SE) in females collected before mating, interrupted at the end of stage I before insemination, and after one complete copulation.** Numbers are sample size. The pictures show examples of the *bursa copulatrix* full of sperm before and after copulation (left and right) and partially emptied at the end of stage I (centre). Scale bar in the images 0.1 mm. Images edited to remove dust.

expensive. This was an unexpected result, because other small non-territorial damselflies do not have elaborate courtships, and suggests that precopulatory sexual selection might be intense in this species. The complexity of this precopulatory courtship makes this species not only unique from its morphology and taxonomic position but also ethologically. The specialized use of the legs to immobilise the female for long periods is not found, to my knowledge, in other damselflies. The complete sequence of precopulatory tandem and courtship might be primitive in comparison with other damselflies.

When small zygopterans are handled for marking, it is not rare that some lose one or more legs, a fact that might reduce their survivorship (*Cordero-Rivera, Egido Pérez & Andrés, 2002*). In the case of *H. mirabilis* this never happened, and all individuals had six legs when first captured. Furthermore, I found that pulling off legs out from adult *H. mirabilis* to sample tissues for DNA, required the use of scissors instead of forceps, because forcibly removing one leg could damage the animal due to muscular tissues remaining attached to the leg. Given that legs are used to immobilise females during courtship (Figs. 1 and 2), perhaps this explains why *H. mirabilis* males do not autotomise their legs. Legs are crucial for male courtship in *H. mirabilis*, another unusual characteristic.

A third peculiarity of *H. mirabilis* is that mating frequency is surprisingly low. Even if males and females were observed in large numbers, they seldom interacted. The rarity of mating interactions seems the norm among Amazonian rainforest odonates (Rhainer

Guillermo, pers. comm, 2016 and also pers. obs.), which have low densities and live for weeks or months, but it is certainly surprising for an animal with a high density and low daily survivorship like *H. mirabilis* (*Cordero-Rivera, 2016*). This low mating frequency, combined with the intense precopulatory sexual selection during courtship and lack of postcopulatory association between males and females, which is typical when polyandry is common (*Alcock, 1994*), suggest that females mate very few times during their lifetime. This is further indicated by the finding that two out of seven pre-copula females and two out of eight females interrupted at the end of stage I of copulation had no sperm from previous matings, which suggests that they were unmated or might be mating for the first time.

Even if mating frequency is low, it is unlikely that female *H. mirabilis* is monogamous. The anatomy of genitalia is typical of sperm removers, and the measurement of sperm volumes confirmed this possibility. Comparison of Figs. 7A and 7B show that the size and morphology of genital ligula match the size and morphology of female genitalia. The images in Fig. 7 are oriented with the dorsal part above, but during copulation, the abdomen of the female contacts with the male upside-down as in Fig. 4. During copulation, the foldable tip of the genital ligula is oriented in a way that favours the introduction of its horns into the *bursa*, facilitating sperm capture and removal. This positioning was confirmed in one pair preserved in copula. In all Zygopteran species so far studied, stage I is used to remove sperm from previous matings and stage II to inseminate (*Córdoba-Aguilar, Uhía & Cordero-Rivera, 2003*). My results are compatible with this scenario also for *H. mirabilis*. Therefore, sperm competition has been a relevant force in the evolution of reproductive behaviour in this species, and given its phylogenetic position in the order Odonata, it may date back to the Permian. A study of the evolution of genitalia in a phylogenetic context is certainly a priority (e.g., *Rudoy & Ribera, 2016*). Two (out of nine) females dissected after a complete copulation did not have sperm in their *bursa*. This suggests that some matings might be unsuccessful at insemination, and therefore females may need to mate more than once to be able to reproduce. This possibility needs further study.

I did not observe oviposition, including trying to elicit egg-laying on humid filter paper and plant tissue. Oviposition has apparently never been observed in this species (*Sant & New, 1988*). The dense vegetation of Long Swamp was too complex to allow detailed behavioural observations of these small damselflies. If females lay eggs at the base of the reeds, this is unlikely to be detected. Even mating pairs were very difficult to observe among the vegetation. Furthermore, individuals in copula were never seen flying (a further peculiarity of this species), which also makes detection difficult. Focal observation of 79 females in this population (each with a duration of 10 min) allowed witnessing three of them mating, but none laying eggs (*Cordero-Rivera, 2016*). One possibility is that oviposition takes place at night. Nevertheless, this seems unlikely because *Hemiphlebia* showed no activity at low temperatures, and five females were confined overnight with humid filter paper, and did not lay eggs. Further detailed observations at localities where the vegetation is less dense (e.g., Ming Ming swamp) might allow one to detect oviposition, which surely is endophytic given the structure of the ovipositor (*Sant & New, 1988*).

To conclude, this study offers the first description of reproductive behaviour of a key taxon in the evolution of the Odonata, considered the sister group to all Lestoidea

(*Dumont, Vierstraete & Vanfleteren, 2010*), and suggests that sperm removal is an old adaptive trait within male odonates in the arena of sexual selection. Some mysteries remain: ''*Hemiphlebia mirabilis* will always be an enigma'' (*Fraser, 1955*).

## ACKNOWLEDGEMENTS

I acknowledge the help of many colleagues who shared their experiences and information about the fascinating *H. mirabilis* during my sabbatical leave in Nov–Dec 2013, with a license from the University of Vigo. Many thanks to Di Crowther, Ian Endersby, Gerry Quinn, John Hawking, Reiner Ritchter, and Richard Rowe. I am grateful to Natalia von Ellenrieder and Rosser W. Garrison for their useful comments.

### Funding
Funding was provided by grants from the Spanish Ministry of Science and Innovation, which included FEDER funds (CGL2011-22629 and CGL2014-53140-P). The funders had no role in study design, data collection and analysis, decision to publish, or preparation of the manuscript.

### Grant Disclosures
The following grant information was disclosed by the author:
Spanish Ministry of Science and Innovation: CGL2011-22629, CGL2014-53140-P.

### Competing Interests
The author declares there are no competing interests.

### Author Contributions
- Adolfo Cordero-Rivera conceived and designed the experiments, performed the experiments, analyzed the data, contributed reagents/materials/analysis tools, wrote the paper, prepared figures and/or tables, reviewed drafts of the paper.

### Field Study Permissions
The following information was supplied relating to field study approvals (i.e., approving body and any reference numbers):
   Victorian Department of Environment and Primary Industries (permit number 10006907).

### Data Availability
   The raw data has been supplied as Supplemental Information 1.

### Supplemental Information
Supplemental information for this article can be found online at http://dx.doi.org/10.7717/peerj.2077#supplemental-information.

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
