# Peer review of "Sperm removal during copulation confirmed in the oldest extant damselfly, Hemiphlebia mirabilis"

_PeerJ, doi:10.7717/peerj.2077_

## Round 0.1 · original submission · Major Revisions

Three reviewers have evaluated your manuscript. Comments are positive. Please revise it accordingly to reviewers' comments, and take care particularly of the suggestions from Reviewer 2.

Kind regards,

Reviewer 1 ·

Basic reporting

The paper is clearly written and structured appropriately. There are a few typos and sentences with an awkward construction which are listed below. Figures are excellent and greatly add to the quality and readability of the paper. The video is also an asset.

Line 111 replace "was" with "were" (genitalia - plural)

Line 122 "reminds that" - awkward construction

Line 125 insert "with" between tandem___ other males

Line 126 delete "are" between "when __ immobilised"

Line 138 should be ..."the male opens its legs widely and releases the female's wings"

Line 140 "performed" (past tense)

Video: typo in caption: strong "wings" should be "winds".

Experimental design

No comment.

Validity of the findings

No comments.

Additional comments

Very interesting!

A drop of sperm is sometimes visible when the male retracts his aedeagus at the end of Stage 1. Was this ever observed?

Is autotomize really the best term for the fragility of damselfly legs or the propensity to lose them? In lizards, tail autotomy is an active defense mechanism. Are you suggesting that losing legs is an adaptation?

The idea that sperm replacement in this species indicates that it evolved early in Odonata is compelling, but perhaps you could speculate further that the entire mating sequence is primitive or plesiomorphic when compared to the behavior ofvother damselflies. How unique is the behavior of clasping the female's wings when the male first takes her into tandem rather than quickly clasping her behind the head? This would seem to be less efficient and more prone to the male losing his grip on the female or to be broken up by another male. On the other hand, the Z-position of the receptive female would seem to enable the male to swing his abdomen forward.

Reviewer 2 ·

Basic reporting

1. The article needs to be edited a bit for issues of English syntax.

2. in Lines 62-64 the author says that research on odonate sperm competition is “confined to a few families”. In fact substantial work has been done on sperm competition in many species of odonates since 1979. Much, but not all of this work is reviewed in Córdoba-Aguilar A., Uhía E., Cordero-Rivera A. 2003 (the paper cited in support of the claim). I think the intent was to justify another study on this particular odonate. That was unnecessary given the uniqueness of H. mirabilis.

Experimental design

The sperm competition analysis is based on Waage 1979. In that study sperm volumes were measured in 4 categories of females:
1) collected on stream (= did most females carry sperm - if so how much?)
2) females in tandem but not yet mating (= did mating females cary sperm? if so how much?)
3) females from interrupted copulations ( = is there less sperm than expected based on females in 1 and 2?)
4) females collected after mating (= how much sperm does female gain from mating? Is there more there than before mating?)

The Cordero-Rivera study only includes a small number of females (n=11) from categories 3 and 4 (3 additional females were exclude for having no sperm at all - 2 after copula and one before). See lines 179-181 in MS and Fig 8. Females had more sperm after mating than during mating.

The results are suggestive (consistent with) sperm removal, but also with the possibility that females only mate if that have no or little sperm (= no sperm removal). Without knowing how much sperm category 1 and 2 females are carrying, there can be no definitive conclusion that sperm removal occurs.

In other words, Waage 1979 showed that sperm volumes of randomly collected and pre copula females were equal and both were equal to sperm volumes in post copula females. Females in copula have significantly less sperm than the other 3 categories. Thus sperm was present to remove and was removed and replaced. It was also clear that the sperm in postcopula females came from the male they copulated with. Critical data in that study are missing from the present one, thus the design is different and does not provide definitive evidence of sperm removal.

The protocol for measuring sperm volumes was the same for the cited paper by Cordero & Miller 1992. In that paper sperm volume comparisons were made, as in Waage 1979, among pre, interrupted and post copula females

Validity of the findings

Given how similar the male and female genitalia are to several species of Calopteryx damselflies that have been definitively shown to remove and replace sperm, and given that the study does have some results at least consistent with sperm removal (but also with mating with sperm depleted females and no removal), it seems reasonable to think that H. mirabilis males do remove sperm. However, I do not think the paper in its present form is acceptable. The core argument of the paper deals with using a supposedly ancient species to see if sperm removal has been present for a long time. That argument requires actually showing removal is or is not going on. Th paper does not do that.

Unless the author has preserved specimens of females collected before mating and in the general population that can be dissected and measured, the paper with its current data and conclusions is not acceptable. If such data do not exist, perhaps a revision can be resubmitted where the data are presented as preliminary but inconclusive and the morphology relative to the existing literature is used to predict sperm removal and suggest how it can be determined. That however greatly weakens the conclusion that this representative of an ancestral condition suggests sperm removal in the context of sperm competition has been present from close to the beginning of odonates.

This is interesting information about a rare and “ancient” species. The behavioral data are intriguing, especially since matings are rare and ovipositions have not been seen. The speculation about what the mating system is goes a bit too far for the data at hand, but a discussion of what it may suggest and the kind of data that needs to be collected to test among possibilities might be useful. That information and suggestions for future work alone, though incomplete, might warrant a note in PeerJ. I think the part about leg atomization can be left out. In my experience handling thousands of individuals across many species, it is rare. Missing legs are certainly rare in nature, but the are such an essential part of obtaining food that it is easy to speculate why on that basis alone.

·

Basic reporting

The author provides a detailed study of the reproductive behavior and the sperm removal ability of a rare and unique damselfly species. H. mirabilis is trully considered a living-fossil, located at the base of Zygoptera phylogeny and probably one of the few representatives of the ancient evolutionary history of Odonata.
Surprisingly, the author adds evidence to the basal position of the sperm removal trait in damselflies, suggesting a possible ancestral state in the group.
The article and all procedures adhere to PeerJ policies.

Minor comments:
1) Ethical statement missing in he Materials and Methods section of the manuscript.

Experimental design

No Comments.

Validity of the findings

Although the author provides a small sample size, the evidence is enough to support his conclusions.

Additional comments

In general, I believe the author provides a detailed study of the reproductive behavior and the sperm removal ability of a rare and unique damselfly species. H. mirabilis is trully considered a living-fossil, located at the base of Zygoptera phylogeny and probably one of the few representatives of the ancient evolutionary history of Odonata.
Surprisingly, the author adds evidence to the basal position of the sperm removal trait in damselflies, suggesting a possible ancestral state in the group.

However, I would like to suggest that the author could expand discussion on the evolution of sperm removal in Odonates in general, and maybe include references of fossil records (if any) and the abscence of such trait in Protodonata. The author should also discuss other possible roles of the abdomen flicking display made by males and females.

---

## Round 0.2 · Minor Revisions

Dear Author,

The ms has been revised carefully, and can now be accepted in PeerJ. However, before final acceptance, please modify the abstract as required by Reviewer 2. Thanks.

Kind regards,

Reviewer 2 ·

Basic reporting

The manuscript has been improved greatly and reads smoothly now.

Experimental design

I fully understand the reluctance to dissect more females of this rare species. However, having added even a few more specimens brings the experimental design into a place where a conclusion can be made. I think the results are now consistent with major sperm removal during copulation in this species.

Validity of the findings

My previous reservations are basically taken care of by adding the missing data on pre-copula sperm volumes. I still feel the leg atomization is intriguing but pushed a bit - seems another reviewer also feels that way. I will defer to the editor on this one.

One final suggestion concerns the last sentence of the abstract:
"These results point out that sperm removal is an old character in the evolution of odonates, probably dating back to the Permian." Simply changing "probably" to "possibly" would bring the conclusion in line with the kinds of evidence available. This is a primitive species in respect to being a member of an ancient linage. There is, however, no evidence that the species itself has existed since the Permian. There is also the possibility that the sperm removal and related morphology represented this species is a recent evolutionary event and the lack of a fossil record of the involved morphology does not allow us to determine when it happened.

This species has genital morphology much more similar to Calopterygids than other Zygoptera and that is intriguing. Do the most recent phylogenetic analyses for odonates put Calopterygids closer to this ancient group or to Lestids with very simple morphology?If this is an ancient species and Calopteryx is more recent - why so similar in morphology compared with other zygopterans? Seems there may be more to examine here before "probable".

Additional comments

Thanks for adding the specimens. We both suspected that removal was there, but there is a need to exclude alternative possibilities. Still very interesting that there were females devoid of sperm and that the genitalia seem so "Calopteryx-like".

---

## Round 0.3 · accepted · Accept

The manuscript can be accepted now.